# First Assessment of Micro-Litter Ingested by Dolphins, Sea Turtles and Monk Seals Found Stranded along the Coasts of Samos Island, Greece

**DOI:** 10.3390/ani12243499

**Published:** 2022-12-11

**Authors:** Guido Pietroluongo, Belén Quintana Martín-Montalvo, Simone Antichi, Anastasia Miliou, Valentina Costa

**Affiliations:** 1Archipelagos Institute of Marine Conservation, P.O. Box 42, 83103 Pythagorio, Samos, Greece; 2Department of Comparative Biomedicine and Food Science, University of Padova, Viale dell’Università 16, 35020 Legnaro, Italy; 3Departamento de Ciencias Marinas y Costeras, Universidad Autónoma de Baja California Sur, Sur KM 5.5, La Paz 23080, Mexico; 4Department of Integrative Marine Ecology, Stazione Zoologica Anton Dohrn (SZN), Contrada Torre Spaccata, Località Torre Spaccata, 87071 Amendolara, Italy

**Keywords:** cetaceans, sea turtles, monk seals, marine contamination, Mediterranean Sea, litter, microplastics

## Abstract

**Simple Summary:**

This study is the first to assess the presence of micro-litter ingested by four species of marine mammals and two species of sea turtles, found stranded along the coastline of Samos Island, north-eastern Aegean Sea, Greece. Litter particles, identified as microplastics, were ubiquitous throughout all the digestive tracts analysed. This study suggests there is widespread microplastic contamination in marine megafauna in the study area, and proposes a method of standardisation to facilitate comparisons.

**Abstract:**

This study is the first to assess the occurrence of micro-litter ingested by marine megafauna in the north-eastern Aegean Sea. A total of 25 specimens from four species of marine mammals, including dolphins and monk seals, and two species of sea turtles, found stranded along the coastline of Samos Island, Greece, were selected for the analysis. Litter particles, identified as microplastics (MPs), were ubiquitous throughout all sections of the gastrointestinal tract (oesophagus, stomach and intestine) in all specimens. Overall, the MPs most frequently found were black fibres 0.21–0.50 mm in size. These results provide insight into the extent of micro-litter ingestion and contamination in marine vertebrates. Here we propose a method of standardisation to establish a baseline for marine taxa in this region of the Mediterranean Sea, where knowledge of the topic is still lacking.

## 1. Introduction

Marine litter is defined as “any persistent, manufactured or processed solid material discarded, disposed or abandoned in the marine coastal environment” [1]. Numerous quantitative reports have illustrated the ubiquity of marine litter, with its presence reaching the remotest islands [2], polar waters [3], and even the deep seafloor [4]. On a global scale, the highest percentage of marine litter consists of plastics [5,6,7,8]. The major impacts of plastics on marine fauna are a result of ingestion [9], entanglement [10] and chemical pollution [11] affecting, among other things, resource acquisition, health and reproductive output [12]. Over the last half-century, along with increasing global plastic production [13], research efforts have also increased, as shown by the rising number of peer-reviewed literature and meta-analyses. Despite recent efforts for the harmonisation of techniques and standardisation of methodologies [14,15,16], different approaches have been employed for the extraction, quantification and identification of plastic debris [17,18,19], hampering comparisons between studies.

The Mediterranean Sea is one of the most impacted marine areas in the world, registering up to 43.55 items of anthropogenic marine litter per 100 m^2^ of seafloor [20]. An estimated 70–80% of Mediterranean marine debris is of plastic origin, of which a large proportion are MPs [21,22]. In the European Union, the Marine Strategy Framework Directive (MSFD-2008/56/EC) aims to establish a “Good Environmental Status” in European seas by considering 11 qualitative descriptors, which include “Biological Diversity” and “Marine Litter”. As defined by the MSFD, a size of 5 mm is the limit used to differentiate mesoplastics (5–25 mm) from microplastics (<5 mm) [23].

The occurrence of MPs has been recorded in the digestive content of a wide range of marine wildlife, including zooplankton [24], fish [25], amphipods [26], sea turtles [27], seals [28] and cetaceans [29,30,31]. These particles can be ingested directly or accumulated through the trophic web [32]. MPs can result in malnutrition in invertebrates [33], impaired reproduction [34] and can also act as vectors of toxins and pathogens [35]. While at lower trophic levels the impact of MPs is more evident [34], their effect on top predators is still poorly understood [31,36]. Among higher trophic levels, indicator species can be considered as “sentinels” to assess ocean health and changes, including the extent of plastic pollution across the marine trophic web [37]. Marine vertebrates, such as cetaceans, sea turtles and pinnipeds, are widely considered as reliable “sentinels” due to their life span, feeding habits and position at the top of the trophic web [27,38,39].

In the north-eastern Aegean Sea, a few studies have reported data on MP pollution in sediments [8] and marine biota [40,41,42], but very little is still known about the occurrence of MP contamination in marine megafauna. Due to the difficulties of investigating the occurrence and effects of MPs in living organisms, research has relied on both living and deceased animals under human care [22]. Over the last decade, research efforts have increased through the analysis of digestive tracts [28,30], including the historical collection of stomach contents [29]. This study aims to provide the first overview of the occurrence and characterization of micro-litter in four species of marine mammals, including dolphins and monk seals, and two species of sea turtles, by analysing the gastrointestinal tracts (GIT: oesophagus, stomach/stomach compartments, small and large intestine) of 25 specimens found stranded along the coastline of Samos Island, Greece.

## 2. Materials and Methods

### 2.1. Study Area

Samos is a Greek island located in the north-eastern Aegean Sea with approximately 140 km of coastline. The island is included in the Central Aegean Important Marine Mammal Area [43] and is confirmed as an important habitat for cetaceans [44] and seals [45]. The Aegean Sea is also listed among the areas of the Eastern Mediterranean considered as critical since it has the largest percentage of oceanic foraging grounds for loggerhead turtles [46]. During the period of 2018–2019, 25 specimens found stranded along the shore of Samos Island were selected for this study: 15 marine turtles, family *Cheloniidae* [9 *Caretta caretta* (Linnaeus, 1758) and 6 *Chelonia mydas* (Linnaeus, 1758)]; 2 Mediterranean monk seals, family *Phocidae* [*Monachus monachus* (Hermann, 1779)]; and 8 cetaceans, family *Delphinidae* [3 *Delphinus delphis* (Linnaeus, 1758), 3 *Stenella coeruleoalba* (Meyen, 1833) and 2 *Tursiops truncatus* (Montagu, 1821)] (Figure 1; Table 1).

### 2.2. Sample Collection

The isolation of the GIT was performed according to international standard protocols [47,48]. Following the protocol proposed by the INDICIT program [49] to study marine litter ingestion by sea turtles as indicator species, the size of all digestive tract sections was recorded. For dolphins, the stomach compartments were isolated and analysed separately (forestomach, main stomach and pyloric stomach). To avoid dispersion of the contents, the tracts were isolated with metal surgical forceps or hemp strings. Each tract was carefully rinsed first from the outside to reduce potential external contamination and then was wrung out from one extremity to the other to collect the content. The hollow of the tract was rinsed using distilled water and the resulting fluid collected. The organ was then opened to inspect the internal appearance and the wall gently scrubbed with a metal spatula, before being rinsed again inside-out with distilled water to collect any remaining content. All samples (content and fluids) were stored in glass jars.

### 2.3. Sample Preparation

Samples were rinsed using distilled water and pre-filtered through two metal sieves (500 and 200 μm) to remove food remains and natural debris (e.g., wood particles, stones, sand and food). All the natural debris and marine litter unable to pass through the sieves were isolated and analysed separately. A saturated saline solution (NaCl, density = 0.7 g/cm^3^) was then added to the final mixture (sieved fluids, 1:1) and left to settle for a minimum of 24 h, allowing all the particles to be resuspended [50].

### 2.4. Filtration

The entire contents of each compartment were analysed in aliquots by mixing 20 mL of supernatant (the settled material was checked with a dissecting microscope) with 20 mL of 30% *w*/*v* hydrogen peroxide (1:1) and 0.1 mL of acetic acid, to dissolve organic matter and carbonate, respectively. The final solution (40 mL) was left to settle before being filtered through a glass fibre filter (GFF: 1.2 µm pore size; 47 mm diameter) with a vacuum pump. After filtration, each GFF was placed in a sterilised glass Petri dish, closed and left to dry.

### 2.5. Particle Identification and Categorisation

Particles retained in each filter were observed using a dissecting microscope (AmScope Compact Multi-Lens Stereo Microscope 20X-40X). Following the method described by Hidalgo-Ruz et al. [50], particles were identified as MPs when no organic structures were observed, a homogenous colour was exhibited and a uniform thickness was visible. In addition, the particles were tested using a hot needle [51], as plastic compounds melt or curl when heated. Therefore, such particles are hereinafter addressed as MPs. When unsure, the particles were not classified as MPs and thus excluded from the analysis. MPs were counted, classified by colour and type (fibre or fragment) (Figure 2), measured (using a micro ruler) and grouped into 5 size categories: ≤0.20 mm; 0.21–0.50 mm; 0.51–1.00 mm; 1.01–2.50 mm; 2.51–5.00 mm.

### 2.6. Reagent and Laboratory Contamination

To minimise laboratory contamination, a separate area of the laboratory was equipped and exclusively dedicated to this study. Metal and glass materials were used with all sampling and filtration equipment and operators wore cotton clothes and plastic-free gloves. All the equipment was rinsed with distilled water and disinfected before use.

The reagents used (distilled water, hydrogen peroxide and acetic acid) were filtered beforehand following the same protocol, and the filter papers were examined to analyse for potential contamination. Tests for airborne contamination were performed using control blanks. The tests (n = 10) were conducted at random and simultaneously with each step of the sampling and analysis procedure and observed before and after manipulation.

### 2.7. Data Analysis

Permutational analysis of variance (PERMANOVA) [52] was used to investigate the distribution of size classes of MPs, expressed as an absolute number. PERMANOVA analyses were based on a Euclidean dissimilarity matrix, calculated after a square-root transformation of the raw data. Each term was tested by 4999 random permutations [52]. SIMPER (Similarity Percentage breakdown) [53] procedure was performed to identify the size classes responsible for the differences. To facilitate comparisons, the number of MPs was standardised to the weight of the corresponding empty compartment and expressed as MPs/g (number of items/weight of the empty compartment). MPs/g values showed a non-normal distribution (Shapiro–Wilk normality test, *p* < 0.05) and a non-parametric Kruskal–Wallis test, followed by a Dunn post hoc test, were conducted to investigate the differences between families, species and compartments, and colours. Mann–Whitney U-tests were performed to compare the occurrence of fibres and fragments for each family.

Statistical analyses were performed using R (4.0.3) in RStudio (version 1.2.1335) using the packages *tidyverse* [54], *dunn.test* [55] and *vegan* [56].

## 3. Results

MPs were found in the GIT of all individuals. A total of 10,639 MPs were found (overall mean ± SD = 425.56 ± 402.08 items in the GIT per specimen). The highest number of MP items were found in a *S. coeruleoalba* (n = 2056) and the lowest in a *C. caretta* (n = 87). Most MPs were fibres (n = 8560; 80.5%), the rest being fragments (n = 2079; 19.5%). The majority of items found ranged from 0.21 to 0.50 mm (n = 4634, 43.6%), with smaller proportions of the sizes 0.51–1.00 and 1.01–2.50 mm (n = 2061, 19.4% and n = 2174, 20.4%, respectively). MPs of sizes ≤ 0.20 and 2.51–5.00 mm each represented about 10% of the total items found (n = 701, 6.59% and n = 1069, 10.0%).

A total of 13 different colours of MPs were identified. Blue and black MPs were predominant, respectively, 34.4% (n = 3662) and 27.5% (n = 2923) (Figure 3). Overall, the most frequently found MPs were black fibres of size 0.21–0.50 mm (n = 1073; 10.1%).

There were differences among MP size classes between families (PERMANOVA, df = 2, MS = 508.01, Pseudo-F = 9.672, P(perm) < 0.05). In particular, the sizes 0.21–0.50 mm and 0.51–1.00 mm contributed to a 70% dissimilarity between *Cheloniidae* and *Delphinidae*, as well as between *Delphinidae* and *Phocidae*, while the sizes ≤ 0.20 mm and 0.51–1.00 mm contributed to a 50% dissimilarity between *Cheloniidae* and *Phocidae* (SIMPER analysis). Regarding the occurrence (MPs/g), there were no significant differences between families (χ^2^ = 5.594, df = 2, *p* > 0.05) and species (χ^2^ = 8.008, df = 5, *p* > 0.05). No difference between compartments was observed in *Delphinidae* (χ^2^ = 10.999, df = 5, *p* > 0.05), *Cheloniidae* (χ^2^ = 7.004, df = 3, *p* > 0.05) and *Phocidae* (χ^2^ = 5.500, df = 3, *p* > 0.05) (Figure 4A). A higher number of fibres were found in *Cheloniidae* (W = 177, df = 1, *p* < 0.05) and *Delphinidae* (W = 57, df = 1, *p* < 0.05), while no significant differences between fibres and fragments were detected in *Phocidae* (W = 4, df = 1, *p* > 0.05) (Figure 4B). Colours of MPs were significantly different in *Cheloniidae* (χ² = 75.463, df = 12, *p* < 0.05) and *Delphinidae* (χ² = 63.735, df = 12, *p* < 0.05), with a higher amount of black and blue items. No differences were found in *Phocidae* (χ² = 14.717, df = 9, *p* > 0.05) (Figure 4C).

### Reagent and Laboratory Contamination

An average of MPs lower than 2 ± 0 was identified in the GFFs used for the contamination controls before and during each step of the sampling and analysis procedures (Appendix A).

## 4. Discussion

To the authors’ knowledge, this study is the first to present litter contamination data in the GIT of multiple marine apex predators found stranded in the north-eastern Aegean Sea. Studies comparing MP contamination between different species of marine megafauna are scarce [31], especially in this area of the Mediterranean Sea [15]. Moreover, the absence of globally recognised standardised methods to process samples and evaluate MP items has hindered quantitative comparisons between studies [15,19].

The presence of litter identified as MPs throughout all GIT compartments confirms the ubiquity of these pollutants in apex predators. Differences in foraging behaviour expose each species to the possibility of ingesting primary or secondary micro-litter, both directly and indirectly. Since opportunistic species, such as *T. truncatus*, *M. monachus* and *C. caretta* mostly feed on fish, cephalopods and invertebrates [57,58,59], the majority of MPs could originate from trophic transfers via the ingestion of contaminated prey [32]. Even the strictly herbivorous diet of *C. mydas* [60] could also provide a source of MPs, as seagrass and macroalgae have recently been described as a sink for plastic debris [61]. However, MPs may also be inhaled at the water-air interface [22]. The potential health effects of MP ingestion have been well discussed in marine mammals and turtles [38]. However, the health implications and the long-term effects of plastic ingestion, due for example to chemical compounds and heavy metal adsorption, are far from defined [62]. In the present study, MPs were recorded in the oesophagus of all individuals, which is a less studied compartment than the stomach and intestine [15,30]. The presence of MPs in the stomach is typically explained by the entrapment role exerted by this organ during the passage of the chyme through the GIT [31]. Identifying MPs in every digestive compartment suggests that these particles can travel through the entire GIT and could eventually be egested in the faeces [63], as previously found in *T. truncatus* [64] and grey seals [32]. However, further research is needed to understand how non-digestible particles such as MPs transit through the GIT, remain in each compartment or enter into the blood circle, assessing the impact on animal health.

The most common sizes of MPs recorded in our study were 0.21–1.00 mm. Synthetic particles < 1.00 mm were identified in all marine turtles analysed by Duncan et al. [65], with higher numbers in the Mediterranean Sea [65]. In our study, MPs < 0.50 mm were overrepresented, and drove the difference between *Cheloniidae* and *Delphinidae*. The high amount of smaller size particles could be related to the breaking down action of the transit and digestive process [64]. The majority of MPs detected in *Cheloniidae* and *Delphinidae* were fibres, as reported in other studies [22,30,31,64,65]. This type of MP has been found in almost every marine habitat around the globe [29,66]. The predominant colours of the detected MPs were blue and black, which frequently dominates the composition of particles ingested by fish and marine mammals [22,29,31].

The analysis of a selection of the most recent studies (2010–2021) assessing MP ingestion in stranded marine vertebrates (Appendix A) [22,29,30,31,42,64,65,67,68,69,70,71,72,73,74,75] highlighted that the range of MPs found was greatly variable, from one single particle in 8 specimens of *C. caretta* in North Carolina, USA [65] to 286 MPs in 26 individuals of the same species in Greece [42], or 1964 MPs in 7 specimens of *T. truncatus* in South Carolina, USA [64]. However, comparing the absolute number of MPs per individual without standardisation could be unreliable as the number may depend on the size and weight of the animal, which is not always easy to determine due to the state of the preservation of the carcass (Pietroluongo, pers. obs.). To overcome the anatomical and physiological differences in the studied species and the problem in measuring different carcass sizes, we propose a standardisation to the weight of the empty compartment (in MPs/g). In our study, with the proposed standardisation, MP occurrence was not different between families, species, and compartments, unlike what could have been interpreted by just looking at the absolute values. We thus recommend the standardisation of the number of MPs to the weight of the empty compartment (in MPs/g) as a practical and solid approach to quantify MPs in the GIT of marine stranded megafauna and a contribution to the harmonisation of the method not only in Greek waters, but also as requested by the international scientific community and regulations [15].

Finally, the procedure adopted to minimise laboratory contamination proved to be effective according to Galgani et al. [7], who stated that the number of MPs identified in the control samples must be lower than 10% of the average found throughout all samples. However, field contamination of stranded specimens due to airborne particles remains a possible constraint, as has occurred in other studies [28]. In this study, the particles were identified as MPs based on the hot needle test [51]. Novel, more environmentally friendly (i.e., use of less hazardous solutions) and cost-efficient methodologies have been recently proposed using density separation [16] and polymer analysis (e.g., SEM, FTIR, Raman spectroscopy) and would help to confirm MP type and origin.

## 5. Conclusions

This study shows that cetaceans, turtles and seals, regularly present in the north-eastern Aegean Sea, are exposed to marine litter contamination. Despite some limitations, such as possible airborne contamination and the lack of polymer identification, results provide an insight into the extent of MP pollution and provide a baseline across taxa in this region of the Mediterranean Sea where knowledge is still lacking. The proposed standardisation to the weight of the empty GIT compartment can reduce variation and facilitate comparisons between studies. A longer timeframe and a better understanding of the ecology of the studied species could support MSFD monitoring efforts, with the aim of providing useful guidelines for their conservation.

## Figures and Tables

**Figure 1 animals-12-03499-f001:**
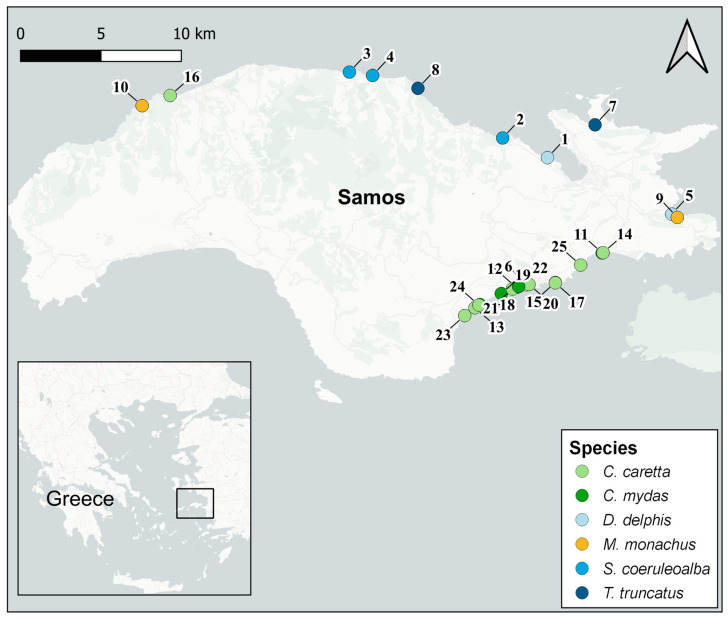
Distribution of stranded individuals along the shore of Samos Island. Point labels correspond to the ID column indicated in Table 1. Map generated using QGIS 3.16.1.

**Figure 2 animals-12-03499-f002:**
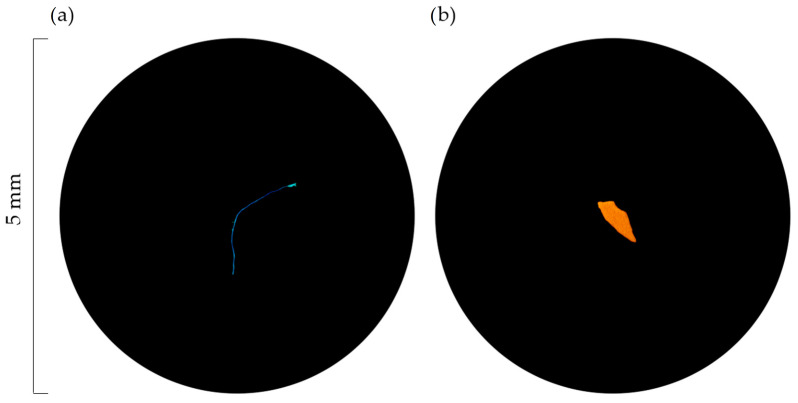
Examples of the types of MPs considered: (**a**) fibre and (**b**) fragment. Image credits: Archipelagos Institute of Marine Conservation.

**Figure 3 animals-12-03499-f003:**
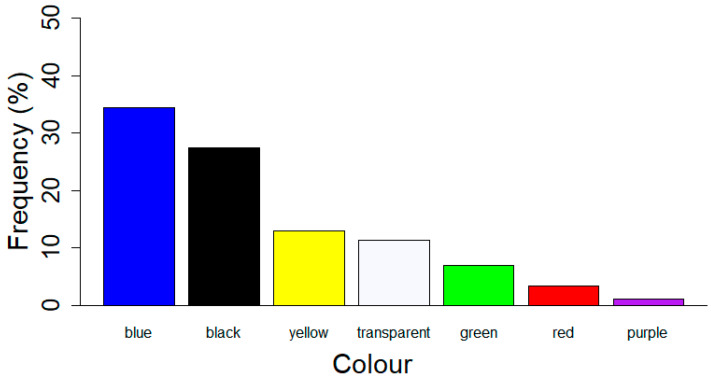
Colours of MPs found expressed as the frequency of occurrence in the entire dataset. Only colours with a proportion higher than 1% are shown.

**Figure 4 animals-12-03499-f004:**
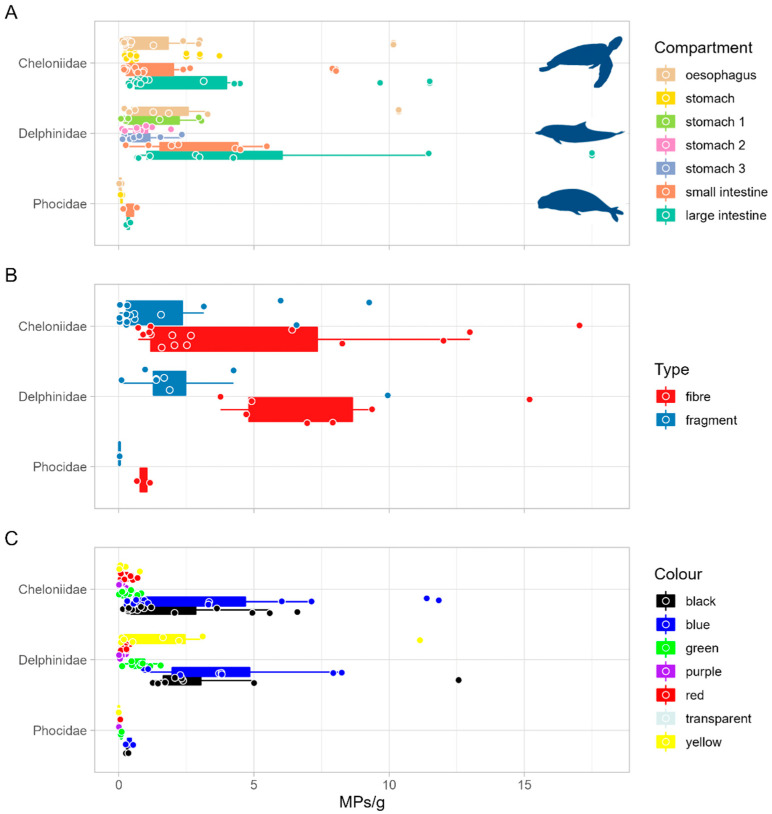
Occurrence of MPs (in MPs/g) per compartment (**A**), type (**B**) and colour (**C**) in the three families analysed. For graphical purposes, only colours with a proportion higher than 1% are shown.

**Table 1 animals-12-03499-t001:** Summary of the specimens found stranded along the coast of Samos Island (Greece) during the period 2018–2019 (F: Female; M: Male; ND: Not Determined; -: Not Measured; MPs: expressed as the raw total number of particles identified in the GIT per individual).

ID	Stranding Date	Lat. (N)	Lon. (E)	Family	Species	Sex	Age Class	Total Length (cm)	MPs
11	12 February2018	37.70749	26.98678	*Cheloniidae*	*Chelonia mydas*	F	Adult	-	133
12	23 February 2018	37.68833	26.92639	*Cheloniidae*	*Chelonia mydas*	ND	Subadult	54	214
13	08 March 2018	37.67701	26.89728	*Cheloniidae*	*Caretta caretta*	F	Adult	110	259
14	10 April 2018	37.70763	26.98737	*Cheloniidae*	*Caretta caretta*	M	Adult	108	237
15	17 April 2018	37.69104	26.95381	*Cheloniidae*	*Chelonia mydas*	F	Adult	85	292
16	22 April 2018	37.79510	26.68289	*Cheloniidae*	*Caretta caretta*	F	Adult	98	296
17	07 November 2018	37.69085	26.95379	*Cheloniidae*	*Caretta caretta*	ND	Juvenile	39	245
18	19 November 2018	37.67877	26.90080	*Cheloniidae*	*Chelonia mydas*	F	Adult	97	503
19	19 November 2018	37.68491	26.91581	*Cheloniidae*	*Chelonia mydas*	F	Adult	84	442
20	22 November 2018	37.69000	26.93533	*Cheloniidae*	*Caretta caretta*	F	Adult	93	165
21	27 November 2018	37.68743	26.92365	*Cheloniidae*	*Caretta caretta*	M	Adult	-	140
22	25 January 2019	37.68881	26.92802	*Cheloniidae*	*Chelonia mydas*	ND	Subadult	53	360
23	30 January 2019	37.67265	26.89008	*Cheloniidae*	*Caretta caretta*	M	Adult	96	287
24	30 January 2019	37.67848	26.90008	*Cheloniidae*	*Caretta caretta*	F	Adult	-	87
25	04 February 2019	37.70077	26.97165	*Cheloniidae*	*Caretta caretta*	F	Adult	93	253
1	09 February 2019	37.76050	26.94826	*Delphinidae*	*Delphinus delphis*	F	Subadult	152	592
2	25 February 2019	37.77147	26.91677	*Delphinidae*	*Stenella coeruleoalba*	F	Juvenile	163	287
3	01 March 2019	37.80810	26.80897	*Delphinidae*	*Stenella coeruleoalba*	F	Subadult	196	773
9	14 March 2019	37.72729	27.03958	*Phocidae*	*Monachus monachus*	F	Adult	205	277
4	16 March 2019	37.80622	26.82526	*Delphinidae*	*Stenella coeruleoalba*	M	Subadult	180	2056
5	24 March 2019	37.72913	27.03561	*Delphinidae*	*Delphinus delphis*	F	Subadult	-	448
6	12 April 2019	37.68975	26.93322	*Delphinidae*	*Delphinus delphis*	M	Subadult	-	402
10	17 April 2019	37.78942	26.66318	*Phocidae*	*Monachus monachus*	M	Adult	260	261
7	22 April 2019	37.77878	26.98176	*Delphinidae*	*Tursiops truncatus*	F	Adult	280	1056
8	04 May 2019	37.79908	26.85716	*Delphinidae*	*Tursiops truncatus*	M	Juvenile	180	574

## Data Availability

The data presented in this study are available on request from the corresponding authors.

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
