# Peer review of "First Assessment of Micro-Litter Ingested by Dolphins, Sea Turtles and Monk Seals Found Stranded along the Coasts of Samos Island, Greece"

_animals, 2022, doi:10.3390/ani12243499_

Round 1
Reviewer 1 Report
This manuscript was well written and clearly presented. Most times when I found myself asking a question, simply reading on answered that question. I appreciate the attention to detail in the methodology to account for contamination. The figures were well constructed and from which it was easy to glean the most salient research points. My only remaining unanswered question was if there was any indication as to the source of the most predominant or frequent fibers (i.e. not where the organisms encountered the plastics, but where they originated) and if their amounts were related to the predominance and frequency of those fibers in the environment.
This is a great contribution to field.
Author Response
R: Dear reviewer, many thanks for your precious help and essential support, it is much appreciated.
Regarding your main question, from our experience in the same area and considering some of the analysis carried out by our group, on the water column and other marine species, the origin of the micro-litter could be related to the textile production line or fishing gear remains. A multidisciplinary approach considering the entire ecosystem coupled with a more specific tracing polymer analysis could support this hypothesis and identify the dispersion and transport behaviours of micro-litter. A future review of all the work conducted in this area could help to improve waste management strategies and/or define a basis for the assessment of the fate and risk of microplastics in this area.
Reviewer 2 Report
The author determined the presence of “micro-litter” meaning microplastics of different sizes and type in stranded marine mammals.
In general the study is well written, short and clear. However I got confused as some points. Based on that I made some suggestions for clarifications of some aspects. See below.
Introduction:
Well-fitting/well introducing the topic
Material and Methods:
Table 1: MPs particles per individual? Maybe not a precise value “per individual”, per g gut volume, … would be more realistic in my point of view.
2.5 line 127/128: provider/company should be mentioned
2.7 line 152: I am not convinced as stated for table 1 on the counting as absolute numbers, there must be a relation to a studied volume
Results:
First paragraph:
please explain more detailed: done counts (per g) or upscaled ? counts per calculate gut volume? Or per animal? (how was it done, what do the values mean? Add to discussion if not done yet!
Figure 3: I think here data in total are shown, should be clarified in figure legend.
I would have liked to see a correlation analysis MPs/g an nutritional behavior (untaken nutrients).
Discussion:
Line 212: “mostly fed on fish” should be specified more precisely for the studies animals
The authors explained different food sources in discussion, line 213: “fish, cephalopods and invertebrates” or line 214/215: herbivorous diet of C. mydas. My comments above exactly refer to those aspects, were differences in MP densities and type determined related to nutritional behavior (see above requested correlation analysis). It would be perfect if the authors good provide something like this!
Line 253: Some sentences should be added here to explain that aspect better, which method is recommended. For me to got not clear.
